# Optimization of In Vitro Regeneration Protocol of Tomato cv. MT1 for Genetic Transformation

**Shiuli Ahmed [1,2], Wan Aina Sakeenah Wan Azizan [1], Md. Abdullah Yousuf Akhond [3], Abdul Shukor Juraimi [4], Siti Izera Ismail [5], Razu Ahmed [6] and Muhammad Asyraf Md Hatta [1,*]**

[1] Department of Agriculture Technology, Faculty of Agriculture, Universiti Putra Malaysia, Serdang 43400, Selangor, Malaysia

[2] Biotechnology Division, Bangladesh Agricultural Research Institute, Gazipur 1701, Bangladesh

[3] Research Wing, Bangladesh Agricultural Research Institute, Gazipur 1701, Bangladesh

[4] Department of Crop Science, Faculty of Agriculture, Universiti Putra Malaysia, Serdang 43400, Selangor, Malaysia

[5] Department of Plant Protection, Faculty of Agriculture, Universiti Putra Malaysia, Serdang 43400, Selangor, Malaysia

[6] Horticulture Research Centre, Bangladesh Agricultural Research Institute, Gazipur 1701, Bangladesh

\* Correspondence: m.asyraf@upm.edu.my

**Abstract:** The tomato (*Solanum lycopersicum* L.) is a major crop of global economic significance. The characterization of genes associated with agriculturally important traits is often performed using genetic transformation. To achieve an efficient transformation protocol, three components are required, namely, a regenerable target tissue, a DNA delivery method, and a robust transformant selection system. The present study was conducted to optimize the in vitro regeneration protocol for the tomato cv. MT1. The regeneration capacity of hypocotyl and cotyledon explants was evaluated using a total of 20 concentration combinations of two plant growth regulators (PGRs) added into the basal MSB5 medium, namely, 6-benzylaminopurine (BAP) (0, 1, 2, 3, and 4 mg/L) and indole-3-acetic acid (IAA) (0, 0.05, 0.1, and 0.5 mg/L). The optimal PGRs combinations for the cotyledons and hypocotyls were MSB5 supplemented with 2 mg/L BAP and 0.5 mg/L IAA and MSB5 supplemented with 2 mg/L BAP and 0.1 mg/L IAA, respectively. To determine the minimum inhibitory concentration (MIC) of kanamycin, eight different concentrations (0, 50, 75, 100, 125, 150, 175, and 200 mg/L) were added to the MSB5 supplemented with 2 mg/L BAP and 0.5 mg/L IAA. The MIC for the cotyledons and hypocotyls were determined to be 50 mg/L and 100 mg/L, respectively.

**Keywords:** tomato; *Solanum lycopersicum*; in vitro regeneration; hypocotyl; cotyledon; kanamycin

## 1. Introduction

The tomato (*Solanum lycopersicum* L.) is one of the most economically important crops and is widely cultivated around the world. In terms of total production, it is the most popular vegetable crop in the world. In 2020, the global production of tomato was 186.82 million metric tons, of which 1.25 million metric tons were produced in Cameron Highlands, Malaysia [1,2]. The crop is regarded as a model plant for dicots and is used for gene functional studies that can be applied to other crops [3,4].

Crop genetic transformation is an important tool for studying the role of genes that govern key traits, including biotic factors, abiotic factors, quality, and yield. There are three major components to achieve an efficient transformation system, namely, a regenerable target tissue, DNA delivery method, and a robust selection method for selecting transformants. *Agrobacterium*–mediated transformation has been the most common method for delivering DNA into the tomato genome [5]. In this approach, hypocotyl [6] and cotyledon [7] have been used as the starting materials and the regeneration efficiency varies among genotypes [8–10].

The tomato cv. MT1 developed by the Malaysian Agricultural Research and Development Institute (MARDI) is grown in lowland areas of Malaysia [11]. Most of the previous studies on regeneration of this cultivar used only cotyledons as explants and zeatin as a plant growth regulator (PGR) [12,13]. 6-Benzylaminopurine (BAP) is cheaper than zeatin. A simple protocol that utilizes other potential explants and alternative growth regulators may reduce the regeneration time and cost and increase its efficiency.

A competent selection system for distinguishing transformed cells from non-transformed cells is required for crop transformation. Typically, antibiotic resistance is utilized for the selection of regenerated putative transformants. For this purpose, antibiotic resistance genes are often incorporated into transformation vector to select plant transformants [14,15]. The neomycin phosphotransferase II (*npt II*) gene, which confers resistance to kanamycin, is the most commonly used selectable marker gene for screening transformants [16]. Through expression of the incorporated antibiotic resistance gene, successful transformants can detoxify a specific antibiotic [16,17]. Thus, the selection of positive transformants is achieved using an antibiotic at the minimum required concentration in the regeneration medium, also known as the selection medium. The toxicity of an antibiotic depends on the organism, species, and concentration [18–20]. Plant regeneration and growth are inhibited at higher concentrations of antibiotics [17,21]. Prior to transformation, the minimum inhibitory concentration (MIC) of the antibiotic for non-transformed plants must be determined to ascertain the level of resistance [22].

In the genetic transformation of tomato cv. MT1, the hygromycin resistance gene is often used as a selectable marker gene [23,24]. Although a study described the use of kanamycin as a selection system in this cultivar [13], no information regarding the determination of the MIC of kanamycin has been documented to date.

Therefore, the current experiments were designed to optimize the regeneration protocol from the cotyledon and hypocotyl explants. The optimal concentration of indole-3-acetic acid (IAA) for rooting media and the MIC of kanamycin for tomato cv. MT1 were determined.

## 2. Materials and Methods

### 2.1. Preparation of Plant Material

Seeds of tomato cv. MT1 were presoaked for 1 h in sterile distilled water. The seeds were then rinsed well with sterile distilled water and soaked in 70% ethanol for 30 s, then rinsed again with sterile distilled water. The seeds were then soaked in a 2% sodium hypochlorite (bleach) solution, added with 25 μL of Tween-20 per 100 mL of bleach solution. The seeds were stirred continuously in the bleach solution for 15 min. The bleach-treated seeds were then rinsed four times with sterile distilled water and germinated on full strength MS (Murashige and Skoog) medium [25]. The explants (cotyledons and hypocotyls) were prepared aseptically from 13–15-day-old seedlings. Cotyledonary nodes with shoot tips and epicotyls were removed. The cotyledons were cut at both ends, maintaining an explant size of 0.5 cm (approximately). The hypocotyls explants were also cut into 0.5 cm (approximately) pieces. Two cotyledons and 2–3 hypocotyl explants were collected from each seedling and placed on the media. In a shoot induction experiment for optimization of the regeneration protocol, the explants from the hypocotyls and cotyledons were referred as H and C, respectively.

### 2.2. Culture Media for Shoot Induction

Murashige and Skoog media, including Gamborg B5 vitamins (MSB5), were used as a basal medium [25,26] supplemented with different concentrations of BAP and IAA. Through a literature search on the regeneration of tomato using BAP and IAA [27], concentrations of BAP in the amount of 0, 1, 2, 3, and 4 mg/L and IAA of 0, 0.05, 0.1, and 0.5 mg/L were selected and used in the current study, respectively. All 20 possible combinations of the BAP and IAA concentrations were used for shoot regeneration (Table 1). For root induction, five concentrations of IAA (0, 0.25, 0.5, 0.75, and 1.0 mg/L) were used with the

MS medium to determine the optimal concentration of IAA for robust and effective root induction.

The combination of MSB5 + 2 mg/L BAP + 0.5 mg/L IAA (T18) was found to be the best medium from the experiment on the optimization of the regeneration protocol. Therefore, it was used as a basal/control medium for subsequent experiments. The regeneration medium was supplemented with different concentrations of kanamycin (0, 50, 75, 100, 125, 150, 175, and 200 mg/L).

**Table 1.** List of treatment combinations of plant growth regulators (PGRs) and explants.

| Treatment | BAP + IAA (mg/L) | Hypocotyls (E1) on Corresponding Treatment | Cotyledons (E2) on Corresponding Treatment |
|---|---|---|---|
| T1 | 0 + 0 | HT1 | CT1 |
| T2 | 1 + 0 | HT2 | CT2 |
| T3 | 2 + 0 | HT3 | CT3 |
| T4 | 3 + 0 | HT4 | CT4 |
| T5 | 4 + 0 | HT5 | CT5 |
| T6 | 0 + 0.05 | HT6 | CT6 |
| T7 | 0 + 0.1 | HT7 | CT7 |
| T8 | 0 + 0.5 | HT8 | CT8 |
| T9 | 1 + 0.05 | HT9 | CT9 |
| T10 | 2 + 0.05 | HT10 | CT10 |
| T11 | 3 + 0.05 | HT11 | CT11 |
| T12 | 4 + 0.05 | HT12 | CT12 |
| T13 | 1 + 0.1 | HT13 | CT13 |
| T14 | 2 + 0.1 | HT14 | CT14 |
| T15 | 3 + 0.1 | HT15 | CT15 |
| T16 | 4 + 0.1 | HT16 | CT16 |
| T17 | 1 + 0.5 | HT17 | CT17 |
| T18 | 2 + 0.5 | HT18 | CT18 |
| T19 | 3 + 0.5 | HT19 | CT19 |
| T20 | 4 + 0.5 | HT20 | CT20 |

H = Hypocotyl, C = Cotyledon.

### 2.3. Culture and Subculture

Both the cotyledon and hypocotyl explants were cultured on MSB5 medium supplemented with different concentration combinations of BAP and IAA. The plates with the explants were incubated at $25 \pm 2\ °C$, maintaining 16/8 h light/dark stages. The explants were sub-cultured every two weeks onto fresh regeneration medium containing the same plant growth regulators (PGRs) combinations. The cultures were maintained using fluorescent light of 20,000–25,000 lux intensity at $25 \pm 2\ °C$ and a cycle of 16/8 (light/dark) hours.

Both the cotyledon and hypocotyl explants were cultured on the best regeneration medium, MSB5 + 2 mg/L BAP + 0.5 mg/L IAA (T18) supplemented with different concentrations of kanamycin. The plates with the explants were incubated at $25 \pm 2\ °C$, maintaining 16/8 h light/dark stages. The explants were sub-cultured every 2 weeks onto similar fresh regeneration medium. Induced shoots were transferred to MS medium for elongation. After two weeks on the elongation medium, the shoots were transferred to root-inducing media. Ten shoots were used per replication for each treatment. The plantlet were kept in the root-inducing media for three weeks.

### 2.4. Shoot Elongation, Rooting, and Acclimatization

Individual shoots > 0.5 cm were excised after regeneration and transferred to fresh medium for elongation. Three types of media (MSB5 + 2 mg/L BAP + 0.1 mg/L IAA, MSB5 + 1 mg/L BAP + 0.1 mg/L IAA, and MSB5) were used for shoot elongation.

Shoots derived from the hypocotyl and cotyledon explants in the optimization of the regeneration protocol experiment were referred to as SH and SC, respectively. The shoots



were transferred to root-inducing media after two weeks on the elongation medium. Five concentrations of IAA (0.0, 0.25, 0.5, 0.75, and 1.0 mg/L) were used in MS medium for the root initiation experiment. Ten shoots were used per replication for each treatment. The plantlets were kept in the root-inducing media for three weeks.

The shoots derived from three types of media (MSB5 + 2 mg/L BAP + 0.5 mg/L IAA, MSB5 + 2 mg/L BAP + 0.5 mg/L IAA + 50 mg/L kanamycin, and MSB5 + 2 mg/L BAP + 0.5 mg/L IAA + 75 mg/L kanamycin) were separated into three groups (S1, S2, and S3, respectively). Three media (T1 = MS, T2 = MS + 0.75 mg/L IAA, and T3 = MS +1 mg/L IAA) were used for root induction on the three types of shoots.

The rooted plantlets were kept in the greenhouse without removal from the rooting medium for five days to adapt to the environment and then transferred to autoclaved soil medium (Nursery King, Tray substrate, 010FV03, The Netherlands). The pots with plantlets were kept covered with transparent plastic bags for one week. After one week, the covers were removed, and the plantlets were kept in the greenhouse. Well-established plants were transferred to an open field three weeks after transfer to the soil medium.

*2.5. Experimental Design, Data Collection, and Statistical Analysis*

The experiment was set up in a completely randomized design (CRD) with four replications; in each replication, 16 explants were inoculated in a sterile petri dish containing the regeneration medium. Each experiment was repeated three times. The average data from those three experiments was used for analysis.

The selected parameters for collecting data were percent (%) of explants producing direct shoots, type of induced organ, number of shoots per explant, days to root, number of roots per shoot, root length (cm), root diameter (cm), and % survival of plants in soil. The data were collected by closely observing the cultures. The types of induced organ were recorded through visual observation. The induced organs with a growing tip were considered to be shoots (S = shoot), and the induced organs with no growing tip were recorded as leaves (L = leaf only); explants that had induced roots (R) with or without a shoot and/or leaf were also recorded. The priority of induced organs by a specific explant in response to a specific growth regulator combination was recorded and indicated with sequence of S, L, and R (RS/SR/SL/SRL). The % of explants producing shoots and number of shoots per explant was counted at the completion of six weeks of culture, when the individual shoots were clearly identifiable with growing shoot tips [27]. Only directly regenerated shoots formed without a callus were counted. The number of roots and the root length and root diameter were counted and measured with a slide caliper after three weeks in the root-induction media. Three weeks after planting, the survival rate (%) of the rooted shoots in the soil was recorded.

The data were analyzed using SAS software version 9.3. A CRD 2-factor analysis was performed to compare the individual and interaction effects of two factors. The type of explant (cotyledon and hypocotyl) was considered one factor and the growth regulator combinations (BAP and IAA) were considered another factor for the regeneration experiment. For the MIC of the kanamycin experiment, the explant was considered as one factor and the concentration of kanamycin was considered another factor. Tukey's HSD test was used to compare means at $p < 0.05$.

## 3. Results and Discussion

### 3.1. Regeneration of Shoot

3.1.1. Percentage of Explants Producing Direct Shoots

Significantly different responses were observed among the different types of explants to different concentration combinations of the growth regulators (Figure 1 and Table 2). The percentage of explants forming direct shoots without forming a callus varied from 0 to 98.50%. The highest percentage (98.50%) of hypocotyls and cotyledons producing shoots was observed on T14 (MSB5 + 2 mg/L BAP + 0.1 mg/L IAA) and T18 (MSB5 + 2 mg/L BAP + 0.5 mg/L IAA), respectively (Figure 1). Shoot regeneration was not observed

on the hypocotyl explants treated with T8 (MSB5 + 0 mg/L BAP + 0.5 mg/L IAA). A similar observation was recorded for the cotyledon explants treated with T1 (basal MSB5 medium with no growth regulators), T6 (MSB5 + 0 mg/L BAP + 0.05 mg/L IAA), T7 (MSB5 + 0 mg/L BAP + 0.1 mg/L IAA), and T8 (MSB5 + 0 mg/L BAP + 0.5 mg/L IAA). Only 31.00% of the hypocotyls produced shoots on T1 (basal MSB5 medium with no growth regulators).

In general, a higher number of shoots was observed in the hypocotyl explants in comparison to the cotyledons. A statistical analysis revealed that both the single and interaction effects were highly significant for the percentage of explants that regenerated shoots. This suggests a significant difference in the shoot regeneration capacity of the hypocotyl and cotyledon explants. Similarly, Billah et al. [28] reported that a higher percentage of hypocotyls regenerated shoots than cotyledons. Zaman et al. [29] discovered a contrasting observation, wherein a greater percentage of shoots was regenerated from cotyledons rather than hypocotyls. In a previous study conducted by Sarker et al. [30], it was demonstrated that the optimal combination for shoot regeneration in tomato plants was MS medium supplemented with 1 mg/L BAP + 0.1 mg/L IAA from cotyledon explants, which is different from the present study and is likely due to varietal response.

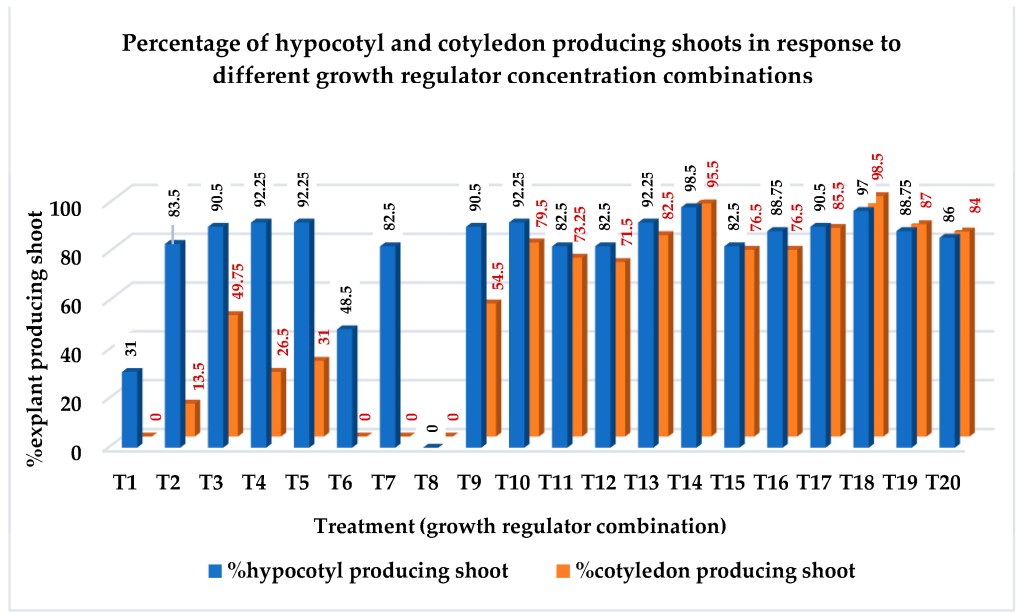

**Figure 1.** Percentage (%) of hypocotyl and cotyledon explants producing shoots on different concentration combinations of BAP and IAA in MSB5 medium.

**Table 2.** Response of MT1 explants to different concentration combinations of BAP and IAA.

| Treatment | % Explants Producing Direct Shoots | % Explants Producing Root | No. of Shoots per Explant | Type of Induced Organ |
|---|---|---|---|---|
| H = Hypocotyl | 79.63 a ± 2.73 | 22.19 a ± 3.97 | 2.73 a ± 0.13 | - |
| C = Cotyledon | 54.28 b ± 3.95 | 17.11 b ± 3.94 | 2.59 a ± 0.19 | - |
| Level of significance | ** | ** | ns | - |
| MSD value | 1.06 | 0.6 | 0.14 | - |
| CV (%) | 5.04 | 9.74 | 17.08 | - |
| HT1 | 31.00 k ± 2.45 | 62.50 b ± 2.55 | 1.00 ef ± 0 | RS |
| HT2 | 83.50 d–g ± 1.50 | 7.81 cd ± 1.56 | 2.25 de ± 0.25 | SR |
| HT3 | 90.50 a–e ± 2.02 | 0 e ± 0 | 2.25 de ± 0.25 | S |
| HT4 | 92.25 a–d ± 1.75 | 0 e ± 0 | 3.25 cd ± 0.25 | SCL |
| HT5 | 92.25 a–d ± 1.75 | 0 e ± 0 | 3.25 cd ± 0.25 | SCL |
| HT6 | 48.50 j ± 1.50 | 100 a ± 0 | 2.25 de ± 0.25 | RS |

**Table 2.** *Cont.*

| Treatment | % Explants Producing Direct Shoots | % Explants Producing Root | No. of Shoots per Explant | Type of Induced Organ |
|---|---|---|---|---|
| HT7 | 82.50 e–h ± 1.50 | 100 a ± 50 | 3.25 cd ± 0.25 | RS |
| HT8 | 0 m ± 0 | 100 a ± 0 | 0.00 f ± 0 | R |
| HT9 | 90.50 a–e ± 2.02 | 6.25 d ± 0 | 3.25 cd ± 0.25 | SR |
| HT10 | 92.25 a–d ± 1.75 | 0 e ± 0 | 2.25 de ± 0.25 | S |
| HT11 | 82.50 e–h ± 1.50 | 0 eh ± 0 | 2.25 de ± 0.25 | LCS |
| HT12 | 82.50 e–h ± 1.50 | 0 e ± 0 | 2.25 de ± 0.25 | LCS |
| HT13 | 92.25 a–d ± 1.75 | 9.38 cd ± 1.80 | 2.25 de ± 0.25 | SRL |
| HT14 | 98.50 a ± 1.50 | 10.94 cd ± 1.56 | 5.25 ab ± 0.25 | SRL |
| HT15 | 82.50 e–h ± 1.50 | 6.25 d ± 0 | 2.25 de ± 0.25 | SRL |
| HT16 | 88.75 b–f ± 1.75 | 6.25 d ± 0 | 3.25 cd ± 0.25 | SRL |
| HT17 | 90.50 a–e ± 2.02 | 12.50 c ± 2.55 | 3.25 cd ± 0.25 | SRL |
| HT18 | 97.00 ab ± 1.73 | 7.81 cd ± 1.56 | 4.25 bc ± 0.25 | SRL |
| HT19 | 88.75 b–f ± 1.75 | 7.81 cd ± 1.56 | 3.25 cd ± 0.25 | SRLC |
| HT20 | 86.00 efg ± 1.00 | 6.25 d ± 0 | 3.25 cd ± 0.25 | SRLC |
| CT1 | 0 m ± 0 | 20.31 c ± 0 | 00 f ± 0 | R |
| CT2 | 13.50 l ± 1.50 | 0 e ± 0 | 1.00 ef ± 0 | S |
| CT3 | 49.75 j ± 2.66 | 0 e ± 0 | 2.25 de ± 0.25 | S |
| CT4 | 26.50 k ± 1.50 | 0 e ± 0 | 2.25 de ± 0.25 | LSC |
| CT5 | 31.00 k ± 2.45 | 0 e ± 0 | 2.25 de ± 0.25 | LSC |
| CT6 | 0 m ± 0 | 100 a ± 0 | 00 f ± 0 | R |
| CT7 | 0 m ± 0 | 100 a ± 0 | 00 f ± 0 | R |
| CT8 | 0 m ± 0 | 100 a ± 0 | 00 f ± 0 | R |
| CT9 | 54.50 j ± 2.87 | 0 e ± 0 | 2.25 de ± 0.25 | SL |
| CT10 | 79.50 f–i ± 1.50 | 0 e ± 0 | 4.25 bc ± 0.25 | SL |
| CT11 | 73.25 hi ± 1.75 | 0 e ± 0 | 3.25 cd ± 0.25 | LS |
| CT12 | 71.50 i ± 2.02 | 0 e ± 0 | 3.25 cd ± 0.25 | LS |
| CT13 | 82.50 e–h ± 1.50 | 6.25 d ± 0 | 2.25 de ± 0.25 | SRL |
| CT14 | 95.50 abc ± 1.50 | 0 e ± 0 | 4.25 bc ± 0.25 | SL |
| CT15 | 76.50 ghi ± 1.50 | 0 e ± 0 | 3.25 cd ± 0.25 | LCS |
| CT16 | 76.50 ghi ± 1.50 | 0 ei ± 0 | 4.25 bc ± 0.25 | LCS |
| CT17 | 85.50 d–g ± 1.50 | 10.94 cd ± 1.56 | 4.25 bc ± 0.25 | SRL |
| CT18 | 98.50 a ± 1.50 | 9.38 cd ± 1.80 | 6.25 a ± 0.25 | SRL |
| CT19 | 87.00 c–f ± 2.44 | 7.81 cd ± 1.56 | 3.25 cd ± 0.25 | LSRC |
| CT20 | 84.00 d–g ± 1.73 | 7.81 cd ± 1.56 | 3.25 cd ± 0.25 | LSRC |
| MSD value | 9.28 | 5.42 | 1.28 | - |
| CV (%) | 4.91 | 9.77 | 17.01 | - |
| Level of (E × T) significance | ** | ** | ** | - |

Means followed by the same letters in a column are not significantly different at a 5% level. MSD = minimum significant difference, CV = coefficient of variation, H = hypocotyl, C = cotyledon. Note: ns—not significant at $p > 0.05$ and **—significant at $p \leq 0.01$; ±—standard error (n = 4); (as ANOVA). C = callus, R = root, S = shoot, L = leaf; the sequence of C, R, S, and L indicates the precedence type of organ.

### 3.1.2. Percentage of Explants Producing Roots

The highest percentage of root formation (100%) in both types of explants was observed in response to three media combinations, i.e., T6, T7, and T8 (T6 = MSB5 + 0 mg/L BAP + 0.05 mg/L IAA, T7 = MSB5 + 0 mg/L BAP + 0.1 mg/L IAA, and T8 = MSB5 + 0 mg/L BAP + 0.5 mg/L IAA), containing IAA only with MSB5 (Table 2). Both the hypocotyl and cotyledon explants produced roots in the basal MSB5 medium without any PGRs, but the percentage was higher for hypocotyl (62.50%) (Table 2). Root induction was inhibited in response to media containing higher concentrations ($\geq 2$ mg/L) of BAP (cytokinin). On average, the percentage of root formation was higher in the hypocotyl explants than in the cotyledon explants. The results reveal that both explants, when cultured on media with no or a low dose (1 mg/L) of cytokinin, produced roots simultaneously with shoots. Similarly, a higher percentage of root formation from hypocotyl explants than from cotyledons was observed by Jamous and Abu-Qaoud [31]. In another study, Jawad et al. [32] observed

simultaneous root formation from explants on MS in combination with cytokinin and auxin.

### 3.1.3. Number of Shoots per Explant

Both single and interaction effects were highly significant for the number of shoots per explant (Table 2). The difference between the hypocotyl and cotyledon explants for the number of shoots produced was clearly observed (Figure 2). The maximum number of shoot regeneration (6.25 ± 0.25) was recorded from the cotyledons on T18 (MSB5 + 2 mg/L BAP + 0.5 mg/L IAA), followed by the hypocotyls (5.25 ± 0.25) on T14 (MSB5 + 2 mg/L BAP + 0.1 mg/L IAA). Billah et al. [28] reported that a similar PGRs combination (2 mg/L BAP + 0.5 mg/L IAA) resulted in the maximum number of shoots per cotyledon explant, whereas, under the same treatment, the hypocotyls formed fewer shoots than the cotyledon explants (Figure 2). The hypocotyls on T18 (MSB5 + 2 mg/L BAP + 0.5 mg/L IAA) and the cotyledons on T10 (MSB5 + 2 mg/L BAP + 0.05 mg/L IAA), T14 (MSB5 + 2 mg/L BAP + 0.1 mg/L IAA), and T16 (MSB5 + 4 mg/L BAP + 0.1 mg/L IAA) produced statistically similar results (4.25 ± 0.25). The hypocotyl explants on T1 (MSB5 + 0 mg/L BAP + 0 mg/L IAA) produced a single shoot per explant. Nevertheless, both types of explants on T8 and cotyledons on T1 (MSB5 + 0 mg/L BAP + 0 mg/L IAA), T6 (MSB5 + 0 mg/L BAP + 0.05 mg/L IAA), and T7 (MSB5 + 0 mg/L BAP + 0.1 mg/L IAA) produced roots but no shoots at all. Similar results were reported by Billah et al., Zaman et al., Ashakiran et al., Jehan and Hassanein, Osman et al., and Gubis et al. [28,29,33–36]. They observed that cotyledon explants produced a higher number of shoots than hypocotyls. Baye et al. [37] found that 2.0 mg/L BAP induced a maximum number of shoots for one of the tomato varieties. A similar result was found by Billah et al. [28] for the effect of the BAP + IAA combination from cotyledon explants. They also discovered that MS medium with 2 mg/L BAP + 0.5 mg/L IAA produced the most shoots per cotyledon explant. Mohamed et al. [38] reported that 2 mg/L BAP gave the maximum number of shoots for both cotyledon and hypocotyl explants.

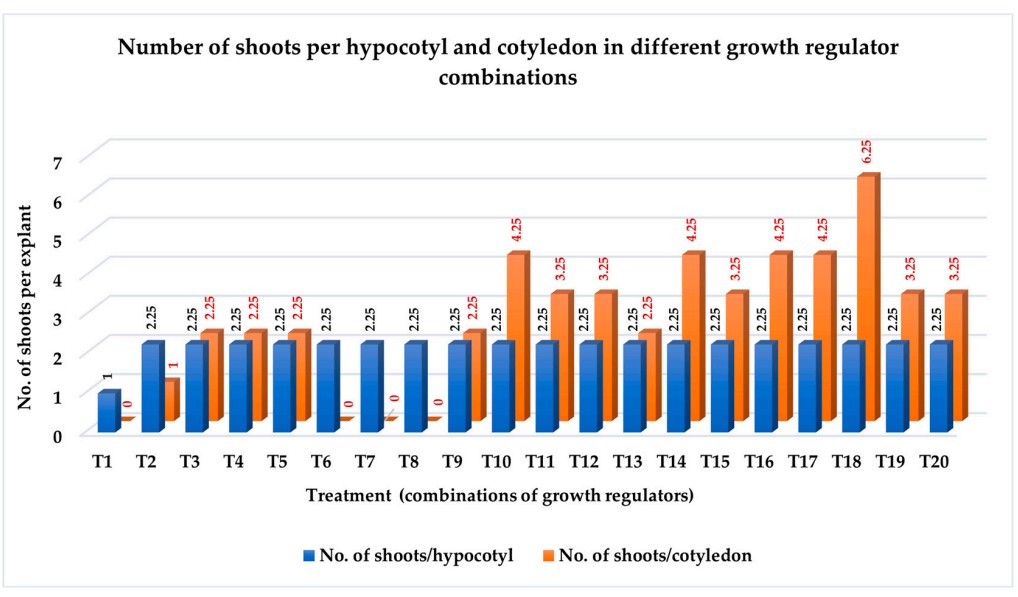

**Figure 2.** Number of shoots per hypocotyl and cotyledon explants on MSB5 supplemented with different concentration combinations of BAP and IAA.

### 3.1.4. Type of Induced Organ

Both the hypocotyl and cotyledon explants showed different types of organogenesis. Shoots, roots, leaves (without growing tips), and calli were observed to grow from the explants. The hypocotyl explants (H) produced shoots on all the media except T8 (MSB5 + 0 mg/L BAP + 0.5 mg/L IAA) (Table 2). The hypocotyl explants produced both roots and

shoots on T1 (MSB5 + 0 mg/L BAP + 0 mg/L IAA), T6 (MSB5 + 0 mg/L BAP + 0.05 mg/L IAA), and T7 (MSB5 + 0 mg/L BAP + 0.1 mg/L IAA), but the roots were more prominent than the shoots (RS) (Table 2). In T11 (MSB5 + 3 mg/L BAP + 0.05 mg/L IAA) and T12 (MSB5 + 4 mg/L BAP + 0.05 mg/L IAA), the hypocotyl explants produced more leaves than shoots (LS) (Table 2). The cotyledon explants on T1 (MSB5 + 0 mg/L BAP + 0 mg/L IAA), T6 (MSB5 + 0 mg/L BAP + 0.05 mg/L IAA), T7 (MSB5 + 0 mg/L BAP + 0.1 mg/L IAA), and T8 (MSB5 + 0 mg/L BAP + 0.5 mg/L IAA) produced only roots (Table 2 and Figure 3c), whereas on T4 (MSB5 + 3 mg/L BAP + 0 mg/L IAA), T5 (MSB5 + 4 mg/L BAP + 0 mg/L IAA), T11 (MSB5 + 3 mg/L BAP + 0.05 mg/L IAA), T12 (MSB5 + 4 mg/L BAP + 0.05 mg/L IAA), T15 (MSB5 + 3 mg/L BAP + 0.1 mg/L IAA), T16 (MSB5 + 4 mg/L BAP + 0.1 mg/L IAA), T19 (MSB5 + 3 mg/L BAP + 0.5 mg/L IAA), and T20 (MSB5 + 4 mg/L BAP + 0.5 mg/L IAA), both the hypocotyl and cotyledon explants produced more leaves and calli (Table 2 and Figure 3d) over shoots (LCS). The hypocotyl explants on T2 (MSB5 + 1 mg/L BAP + 0 mg/L IAA), T3 (MSB5 + 2 mg/L BAP + 0 mg/L IAA), T9 (MSB5 + 1 mg/L BAP + 0.05 mg/L IAA), T10 (MSB5 + 2 mg/L BAP + 0.05 mg/L IAA), T13 (MSB5 + 1 mg/L BAP + 0.1 mg/L IAA), and T20 (MSB5 + 4 mg/L BAP + 0.5 mg/L IAA) and the cotyledon explants on T9 (MSB5 + 1 mg/L BAP + 0.05 mg/L IAA), T10 (MSB5 + 2 mg/L BAP + 0.05 mg/L IAA), T13 (MSB5 + 1 mg/L BAP + 0.1 mg/L IAA), T14 (MSB5 + 2 mg/L BAP + 0.1 mg/L IAA), T17 (MSB5 + 1 mg/L BAP + 0.5 mg/L IAA), and T18 (MSB5 + 2 mg/L BAP + 0.5 mg/L IAA) produced more shoots and fewer roots and leaves (Table 2 and Figure 3e–h).

These results indicate that the cotyledons on the MSB5 media without cytokinin (BAP) and with or without auxin (IAA) produced only roots but no shoots or leaves, whereas the hypocotyl explants produced more roots than shoots without any PGR and with a low concentration (0.05–0.2 mg/L) of auxin only. Both the cotyledon and hypocotyl explants produced vigorous roots but failed to produce shoots on medium containing a relatively higher concentration of auxin (0.5 mg/L IAA) without cytokinin (BAP). Jawad et al. [39] found simultaneous root and shoot formation in response to a combination of higher cytokinin and low auxin. They also observed explants with no shoots for some PGRs combinations.

Explants with higher concentrations (3–4 mg/L) of BAP (cytokinin) produced more calli and leaves without growing tips. In response to lower concentrations (1–2 mg/L) of BAP, with/without IAA, more shoots were produced over roots and leaves. Cotyledons, as compared to hypocotyls, are more likely to produce leaves in response to higher concentrations (3–4 mg/L) of BAP with/without IAA. Jawad et al. [39] reported callus formation on hypocotyl and nodal explants in response to 2 mg/L BAP + 0.5–1 mg/L NAA and 3 mg/L BAP, while Zaman et al. [29] observed a higher rate of callus formation from both hypocotyl and cotyledon explants in response to higher concentrations of cytokinin with low auxin (3 mg/L BAP + 0.2 mg/L NAA). Another study reported that higher concentrations of BAP did not result in shoot regeneration, while callus formation was observed at a BAP concentration of 4 mg/L [38]. The combination of 2 mg/L BAP and 1.75 mg/L kinetin was found to induce a healthy callus and an entire shoot formation, as reported by Hanur and Krishnareddy [40]. The induction of shoots was observed by Jehan and Hassanein [34] with a concentration of 1–2.5 mg/L BAP, either alone or in combination with 0.5 mg/L NAA. They also discovered that 1 mg/L of IAA, IBA, and NAA did not stimulate root formation.

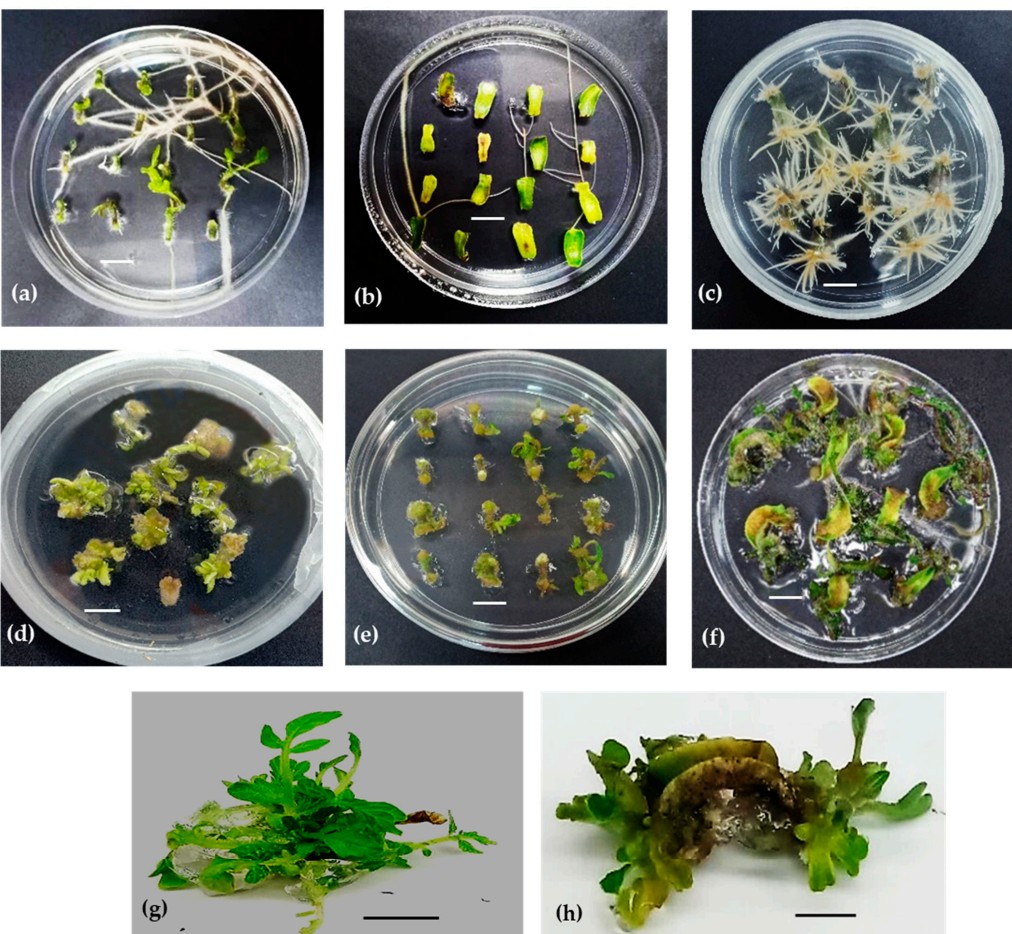

**Figure 3.** Regeneration of shoots from cotyledon and hypocotyl explants of tomato cv. MT1 under different concentration combinations of BAP and IAA: (**a**) HT1 induced roots and shoots (scale bar: 1 cm); (**b**) CT1 induced only roots (scale bar: 1 cm); (**c**) CT8 induced only roots (scale bar: 1 cm); (**d**) E1T4 induced more leaves than calli than shoot (scale bar: 1 cm); (**e**) HT14 induced multiple shoots, roots, and leaves (scale bar: 1 cm); (**f**) CT18 induced multiple shoots, roots, and leaves (scale bar: 1 cm); (**g**) multiple shoots regenerated from hypocotyls (scale bar, 0.5 cm); (**h**) multiple shoots regenerated from cotyledons (scale bar, 0.5 cm). H = hypocotyls, C = cotyledons, T1 = MSB5 + 0 mg/L BAP + 0 mg/L IAA, T4 = MSB5 + 3 mg/L BAP + 0.0 mg/L IAA, T8 = MSB5 + 0 mg/L BAP + 0.5 mg/L IAA, T14 = MSB5 + 2 mg/L BAP + 0.1 mg/L IAA, T18 = MSB5 + 2 mg/L BAP + 0.5 mg/L IAA.

3.1.5. Shoot Elongation

The regenerated shoots were initially transferred onto fresh elongation medium containing MS + 2 mg/L BAP + 0.1 mg/L IAA. The PGR combination resulted in the formation of a callus on the leaves and/or near the shoot tips (Figure 4a). This observation persisted even when the concentration of BAP was decreased within the fresh medium (MS + 1 mg/L BAP + 0.1 mg/L IAA). The observed callus formation on the elongation media might be due to the sensitivity of the regenerated shoots from cv. MT1 towards the PGRs combinations, which accelerated the cell division and dedifferentiation of cells, leading to the formation of calli. Subsequently, shoot elongation was carried out using MS with no PGRs, as the shoots cultured on PGR-free medium exhibited normal elongation (Figure 4c). Cruz-Mendivil et al. and Brassard et al. [27,41] achieved successful and improved shoot elongation of regenerated shoots by using MS media without PGRs.

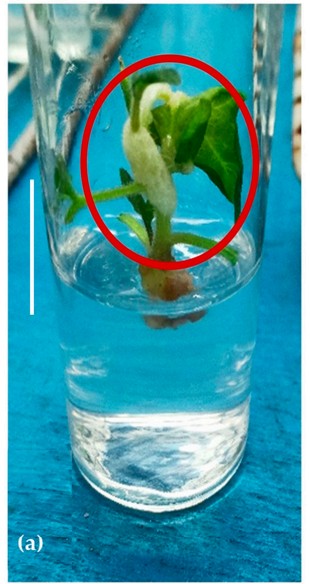
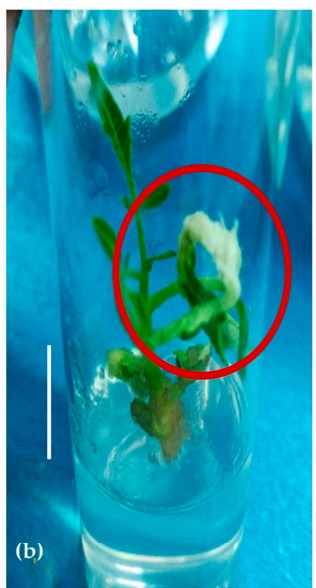
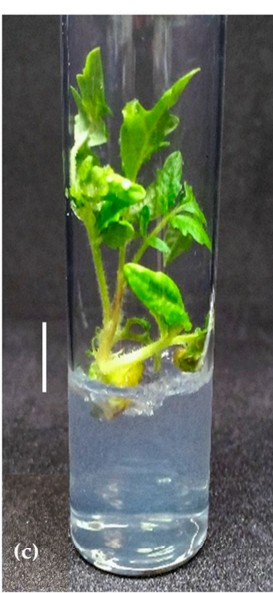

**Figure 4.** Elongation of regenerated shoots. (**a**) Callus (near the shoot tip) growing on the shoot in MS +2 mg/L BAP +0.1 mg/L IAA; (**b**) callus (on leaves) growing on shoot MS +2 mg/L BAP +0.1 mg/L IAA; (**c**) normal growth of shoots on MS medium without PGR. Scale bar = 2 cm.

### 3.2. Root Initiation

The root formation varied significantly between the shoots regenerated from the cotyledon and hypocotyl explants in response to different IAA concentrations (Table 3). Significant interaction effects were observed between the shoots derived from the two types of explants, which were treated with different concentrations of IAA with regard to the duration of days required for root initiation, the percentage of shoots that initiated roots, and the number of roots per shoot (Table 3). However, the interaction effects were not significant for root length, root diameter, and the survival rate of the rooted shoots.

**Table 3.** Effect of different concentrations of IAA on root induction on regenerated shoots from cotyledon and hypocotyl explants.

| Treatment | % Shoot Induced Root | No. of Roots per Shoot | Root Length (cm) | Root Diameter (cm) | Survival Rate in Soil (%) |
|---|---|---|---|---|---|
| SH = Shoot from hypocotyl | 97.50 a ± 0.02 | 20.45 a ± 2.50 | 6.70 a ± 0.056 | 0.38 a ± 0.03 | 60.00 a ± 6.28 |
| SC = Shoot from cotyledon | 61.50 b ± 0.02 | 8.35 b ± 1.16 | 5.41 b ± 0.27 | 0.34 b ± 0.03 | 55.00 b ± 6.75 |
| Level of significance | ** | ** | ** | * | * |
| MSD value | 2.89 | 1.58 | 0.9 | 0.03 | 3.99 |
| CV (%) | 5.62 | 16.96 | 23.11 | 15.94 | 10.77 |
| SH I1 | 87.50 b ± 2.50 | 3.75 e ± 1.11 | 5.75 b ± 0.85 | 0.20 e ± 0.04 | 12.50 c ± 2.50 |
| SH I2 | 100 a ± 00 | 15.00 c ± 1.47 | 5.88 b ± 0.43 | 0.35 cd ± 0.03 | 47.50 b ± 2.50 |
| SH I3 | 100 a ± 00 | 20.75 b ± 1.75 | 10.50 a ± 1.04 | 0.35 cd ± 0.03 | 75.00 a ± 2.88 |
| SH I4 | 100 a ± 00 | 34.00 a ± 1.68 | 5.13 b ± 1.01 | 0.43 bc ± 0.03 | 82.50 a ± 2.50 |
| SH I5 | 100 a ± 00 | 28.75 a ± 1.49 | 6.25 b ± 0.85 | 0.58 a ± 0.05 | 82.50 a ± 2.50 |
| SC I1 | 27.50 e ± 2.50 | 1.50 e ± 0.29 | 5.25 b ± 0.32 | 0.18 e ± 0.03 | 5.00 c ± 2.88 |
| SC I2 | 47.00 d ± 2.50 | 4.00 e ± 0.91 | 5.13 b ± 0.43 | 0.28 de ± 0.03 | 40.00 b ± 4.08 |
| SC I3 | 65.00 c ± 2.88 | 9.50 d ± 0.96 | 7.06 ab ± 0.41 | 0.30 cde ± 0 | 75.00 a ± 2.88 |
| SC I4 | 80.00 b ± 4.08 | 14.00 cd ± 0.91 | 4.88 b ± 0.69 | 0.40 bcd ± 0 | 77.50 a ± 2.50 |
| SC I5 | 87.50 b ± 2.50 | 12.75 cd ± 0.85 | 4.75 b ± 0.48 | 0.53 ab ± 0.03 | 77.50 a ± 4.78 |

**Table 3.** *Cont*.

| Treatment | % Shoot Induced Root | No. of Roots per Shoot | Root Length (cm) | Root Diameter (cm) | Survival Rate in Soil (%) |
|---|---|---|---|---|---|
| MSD value | 11.18 | 5.47 | 3.53 | 0.14 | 14.35 |
| CV (%) | 5.78 | 15.6 | 23.95 | 16.21 | 10.26 |
| Interaction level (S × I) of significance | ** | ** | ** | ** | ** |

Means in a column that include the same letters are not statistically different at a 5% level using Tukey's HSD test. MSD = minimum significant difference, CV = coefficient of variation, I1= MS + 0 mg/L IAA (control), I2 = MS + 0.25 mg/L IAA, I3 = MS + 0.50 mg/L IAA, I4 = MS + 0.75 mg/L IAA, I5 = MS + 1.00 mg/L IAA. Note: ns—not significant at $p > 0.05$, *—significant at $p \leq 0.05$, and **—significant at $p \leq 0.01$; ±—standard error (n = 4); (as ANOVA).

3.2.1. Percentage of Shoots Producing Roots

All the shoots derived from the hypocotyl explants (100%) initiated roots when treated with IAA, whereas the shoots from the cotyledons differed significantly in response to different concentrations of IAA (Table 3 and Figure 5). The root initiation of cotyledon-derived shoots in response to higher concentrations of IAA (I4 = MS + 0.75 mg/L IAA and I5 = MS + 1 mg/L IAA) was similar to the response of the hypocotyl-derived shoots in media without IAA (I1 = MS + 0.0 mg/L IAA). The percentage of rooted shoots was significantly lower for cotyledon-derived shoots (SC) than for shoots derived from the hypocotyl explants (SH). Jawad et al. [39] reported 100% root formation on regenerated adventitious shoots on MS medium supplemented with IAA, which is similar to the present study. Sarker et al. [30] observed maximum root initiation on MS medium with 0.2 mg/L IAA, while Arulananthu et al. [42] were successful in inducing roots using MS medium without PGR. In the present study, although the roots were initiated on MS medium without PGR, the percentage of shoots producing roots was lower than those on MS media supplemented with IAA.

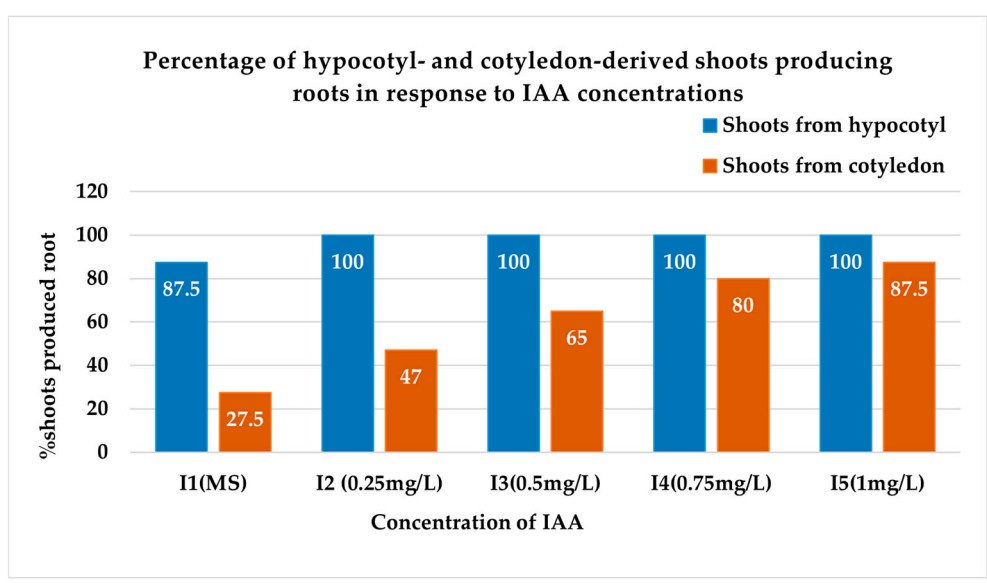

**Figure 5.** Percentage of hypocotyl- and cotyledon-derived shoots producing roots on MS media supplemented with different IAA concentrations.

3.2.2. Number of Roots per Shoot

The hypocotyl shoots (SH) produced the highest number of roots (34 ± 1.68) for I4 (0.75 mg/L IAA with basal MS medium); thus, the interaction of SHI4 was the best for the number of roots per shoot (Table 3). For both SH and SC, the lowest number of roots per shoot was observed for basal MSB5 medium with no IAA (I1). With an increase in the IAA

concentration (I2 < I3 < I4), the number of roots increased, but for I5 (MS + 1.0 mg/L IAA), the number of roots per shoot decreased for both types of shoots (Table 3 and Figure 6). A higher number of roots was observed on the hypocotyl shoots than the cotyledonary shoots.

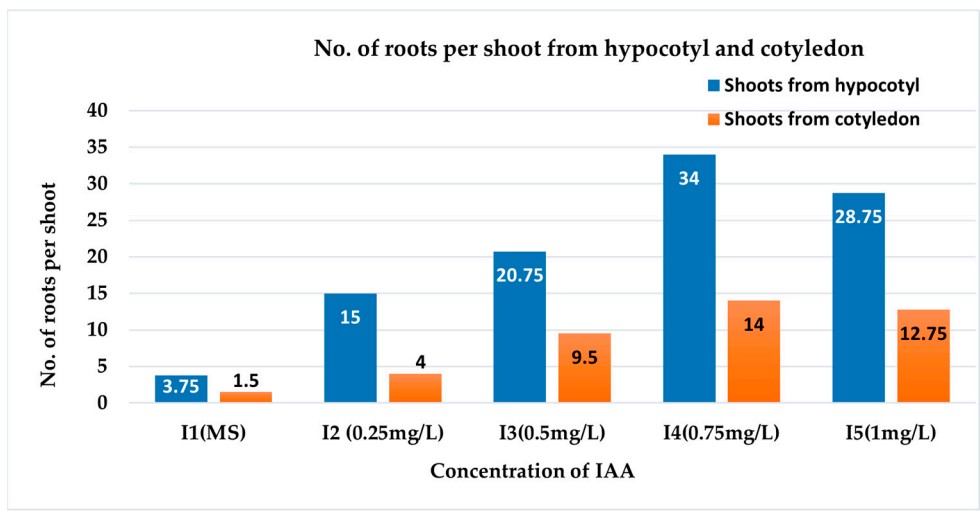

**Figure 6.** Number of roots per shoot derived from hypocotyl and cotyledon explants on MS media supplemented with different IAA concentrations.

### 3.2.3. Root Length and Diameter (cm)

The roots with the longest length (10.50 ± 1.04 cm) were observed as a result of exposure to I3 (MS + 0.5 mg/L IAA) (Table 3). The findings indicate that there was a positive correlation between the concentration of IAA and the length of the root, as evidenced by the increase in root length up to I3 (MS + 0.5 mg/L IAA) (Figure 7c). However, a reduction in the root length was observed again in the cases of I4 (MS + 0.75 mg/L IAA) and I5 (MS + 1.0 mg/L IAA) (Table 3). The study found no statistically significant variation in root length across the various shoot types. The experimental results indicate that the treatment with SHI5 (MS + 1.0 mg/L IAA) resulted in the largest diameter of the roots (0.58 ± 0.05 cm), whereas the smallest root diameter (0.18 ± 0.03 cm) was recorded in the SCI1 (MS + 0.0 mg/L IAA) treatment. There was a positive correlation between the concentration of IAA in the media and the root diameter of both types of shoots. However, it is notable that the observed differences were not statistically significant.

### 3.2.4. Survival Rates (%) of Rooted Shoots in Soil

The percentage of shoot survival in soil varied between 5.00 ± 2.88 and 82.50 ± 2.50% (Table 3). The survival rates of the hypocotyl and cotyledonary regenerants were higher for shoots rooted in I4 (MS + 0.75 mg/L) and I5 (MS + 100 mg/L), as compared to the other treatments. Specifically, the highest survival rate was observed in hypocotyl regenerants rooted in both I4 and I5, with a percentage of 82.50 ± 2.50%. The cotyledonary regenerants rooted in I4 and I5 exhibited a survival rate of 77.50 ± 2.50% and 77.50 ± 4.50%, respectively (Table 3). The present study also discovered an increased number and diameter of roots in response to the application of the two IAA concentrations (I4 and I5). These findings indicate that the survival rates of the regenerants in soil were influenced by the number and diameter of the roots. Previous studies have reported a varied survival rate of healthy rooted regenerants in soil. Yesmin et al. [43] reported a survival rate of 97% for rooted regenerated shoots, and Zaman et al. [29] documented a 100% survival rate. However, Jawad et al. [39] observed a lower success rate of 60% in the establishment of plantlets in soil.

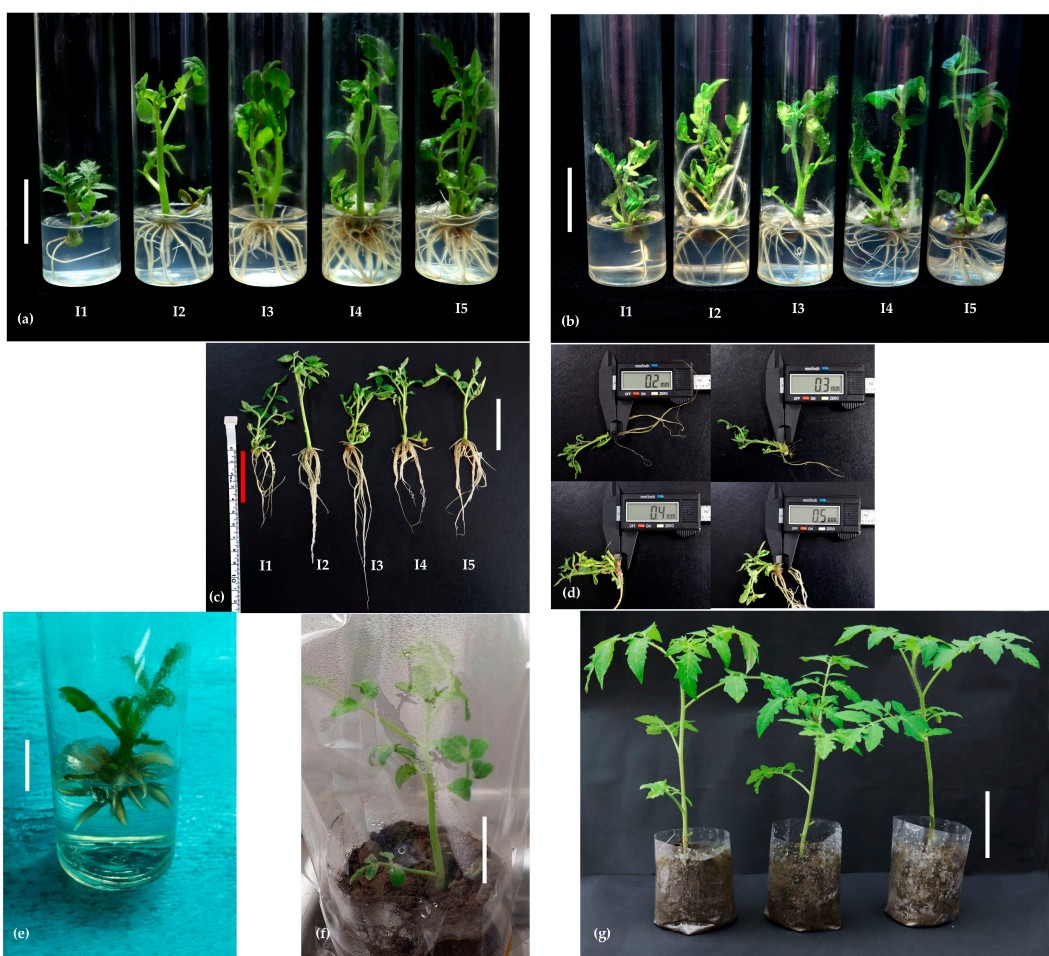

**Figure 7.** Initiation of roots in shoots derived from cotyledon and hypocotyl explants of tomato cv. MT1. (**a**) Rooted hypocotyl shoots (SH) (scale bar: 4 cm); (**b**) rooted cotyledonary shoots (SC) (scale bar: 4 cm); (**c**) root length from shoots treated with different concentrations of IAA (scale bar: 4 cm); (**d**) root diameter from shoots treated with different concentrations of IAA; (**e**) thick root formed on medium I5 (MS + 1.0 mg/L IAA) (scale bar: 2 cm); (**f**) acclimatization of regenerated and rooted plants (scale bar: 4 cm); (**g**) surviving plants (scale bar: 4 cm). I1 = MS + 0 mg/L, I2 = MS + 0.25 mg/L, I3 = MS + 0.50 mg/L, I4 = MS + 0.75 mg/L, I5 = MS + 1.00 mg/L IAA.

### 3.3. Minimal Inhibitory Concentration of Kanamycin on Hypocotyl and Cotyledon Explants
#### 3.3.1. Percentage of Explants Forming Shoots

The regeneration medium (MSB5 + 2 mg/L BAP + 0.5 mg/L IAA) without kanamycin resulted in the highest percentage of hypocotyl and cotyledon explants forming shoots, with a value of 98.25 + 1.75% and 98.44 + 1.56%, respectively (Figures 8 and 9, Table 4). The results also indicate that the addition of kanamycin at concentrations of 50 mg/L and 75 mg/L to the regeneration media led to direct shoot production in the hypocotyl explants at the rates of 48.44% and 13.88%, respectively (Figure 8 and Table 4). The percentage of hypocotyl explants forming shoots exhibited a significant decline in the medium with 75 mg/L kanamycin and was eventually suppressed at 100 mg/L. These findings were comparable with previous studies that used cv. MT1 for tomato transformation [13,44]. In a study conducted by Stavridou et al. [45], a concentration of 100 mg/L kanamycin was employed to inhibit the regeneration of non-transformed shoots in hybrid tomato cultivars Felina, Siena, and Don Jose. In another study, the screening of transformants of various tomato cultivars, including Moneymaker, Pusa Ruby, and Jinan, was performed using 50 mg/L kanamycin [32]. The inhibitory concentrations of kanamycin were observed to vary between 80–200 mg/L for different tomato cultivars, such as Zhongshu No. 5,

Daniela 144, Brillante 179, Annan 3017, Galina 3019, and Bernadine 5656 [44,46,47]. These findings suggest that the optimal concentration for selecting transformants depends on the genotypes of the tomato cultivars.

**Table 4.** Response of tomato cv. MT1 explants to different kanamycin concentrations.

| Treatment | % Explants Forming Shoots | No. of Shoots per Explant |
|---|---|---|
| E1 = Hypocotyl | 20.07 a ± 6.03 | 1.03 a ± 0.30 |
| E2 = Cotyledon | 12.30 b ± 5.85 | 0.81 b ± 0.39 |
| Level of significance | ** | ** |
| MSD value | 1.02 | 0.21 |
| CV (%) | 12.48 | 46.32 |
| T1 | 98.34 a ± 1.09 | 5.63 a ± 0.50 |
| T2 | 24.22 b ± 9.26 | 1.25 b ± 0.50 |
| T3 | 6.94 c ± 2.70 | 0.50 c ± 0.19 |
| T4 | 0 d ± 0 | 0 c ± 0 |
| T5 | 0 d ±0 | 0 c ± 0 |
| T6 | 0 d ±0 | 0 c ± 0 |
| T7 | 0 d ±0 | 0 c ± 0 |
| T8 | 0 d ±0 | 0 c ± 0 |
| Level of significance | ** | ** |
| MSD value | 3.2 | 0.67 |
| CV (%) | 12.48 | 46.32 |
| E1T1 | 98.25 a ± 1.75 | 4.75 b ± 0.48 |
| E1T2 | 48.44 b ± 2.99 | 2.50 c ± 0.29 |
| E1T3 | 13.88 c ± 1.38 | 1.00 d ± 0 |
| E1T4 | 0 d ± 0 | 0 d ± 0 |
| E1T5 | 0 d ± 0 | 0 d ± 0 |
| E1T6 | 0 d ± 0 | 0 d ± 0 |
| E1T7 | 0 d ± 0 | 0 d ± 0 |
| E1T8 | 0 d ± 0 | 0 d ± 0 |
| E2T1 | 98.44 a ± 1.56 | 6.50 a ± 0.65 |
| E2T2 | 0 d ± 0 | 0 d ± 0 |
| E2T3 | 0 d ± 0 | 0 d ± 0 |
| E2T4 | 0 d ± 0 | 0 d ± 0 |
| E2T5 | 0 d ± 0 | 0 d ± 0 |
| E2T6 | 0 d ± 0 | 0 d ± 0 |
| E2T7 | 0 d ± 0 | 0 d ± 0 |
| E2T8 | 0 d ± 0 | 0 d ± 0 |
| MSD value | 5.24 | 1.09 |
| CV (%) | 12.63 | 46.32 |
| Interaction level (E × T) of significance | ** | ** |

Means in a column that include the same letters are not statistically different at a 5% level using Tukey's HSD test. MSD = minimum significant difference, CV = coefficient of variation, T1 = 0, T2 = 50, T3 = 75, T4 = 100, T5 = 125, T6 = 150, T7 = 175, and T8 = 200 mg/L of kanamycin. Note: ns—not significant at $p > 0.05$, and **—significant at $p \leq 0.01$; ±—standard error (n = 4) (as ANOVA).

In contrast, the regeneration of shoots was not observed in the cotyledon explants, except in the regeneration medium without kanamycin, which served as a control (Figure 8 and Table 4). In addition, the occurrence of yellowing was observed on the cotyledon explants (Figure 9b). A similar observation was reported by Wang et al. [17] at a high concentration of kanamycin. Chen et al. [48] reported the observation of leaf yellowing in Arabidopsis explants cultured on kanamycin-containing media. According to Wang et al. (2015) [17], the yellowing and subsequent death of cells in green organs are caused by an interference of kanamycin with protein synthesis in the chloroplasts and mitochondria of plant cells that do not possess the detoxifying ability of the kanamycin resistance gene *nptII*. Duan et al. [49] also suggested that kanamycin affects the growth of cotyledon seedlings and leaves by preventing protein synthesis. Thus, the regeneration of shoots was also inhibited by the restriction of protein synthesis.

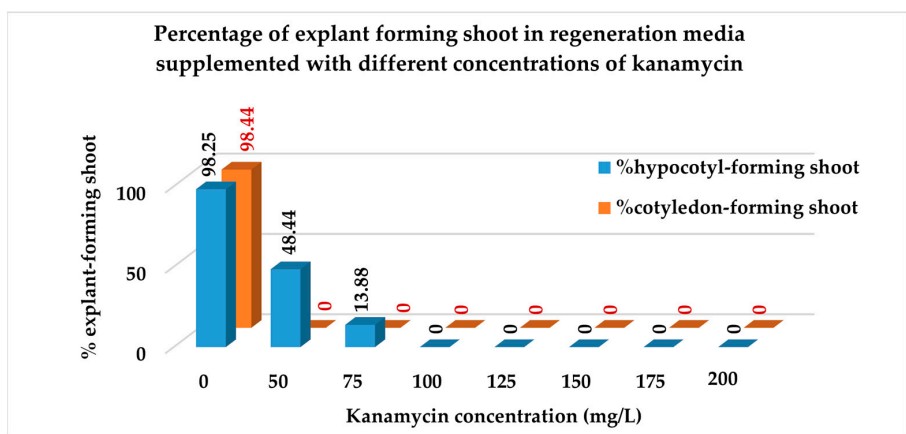

**Figure 8.** Percentage of explants forming shoots on regeneration media supplemented with different concentrations of kanamycin.

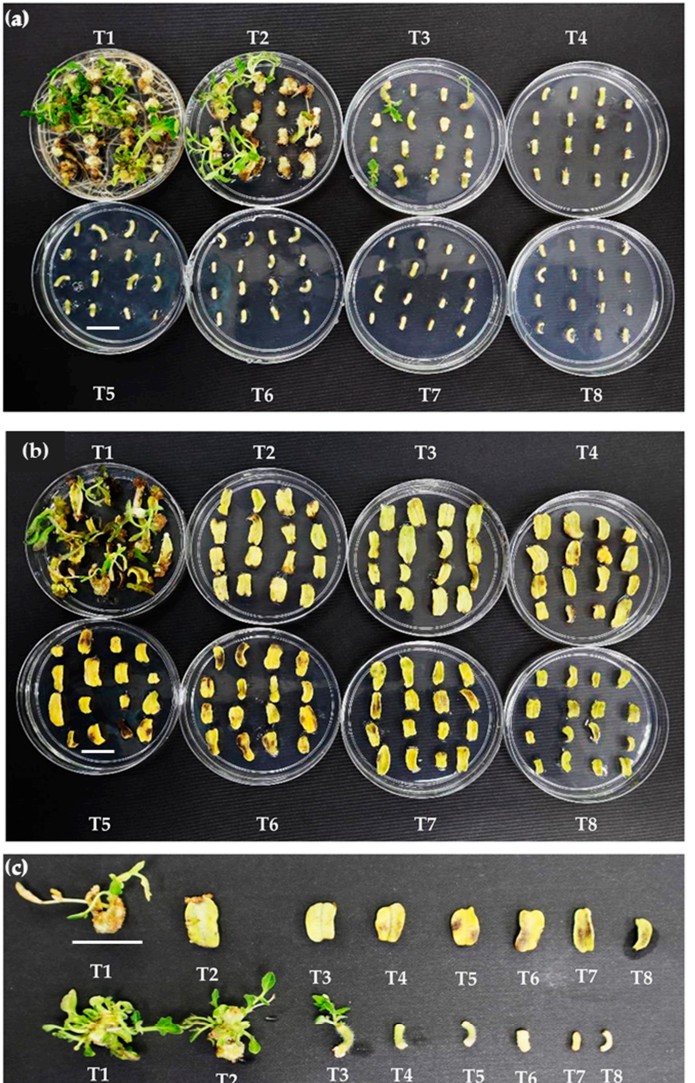

**Figure 9.** Response of tomato cv. MT1 explants to different kanamycin concentrations. (**a**) Response of hypocotyl explants; (**b**) response of cotyledon explants; (**c**) representative explants forming shoots. T1 = 0 mg/L, T2 = 50 mg/L, T3 = 75 mg/L, T4 = 100 mg/L, T5 = 125 mg/L, T6 = 150 mg/L, T7 = 175 mg/L, and T8 = 200 mg/L of kanamycin. Scale bar = 2 cm.

### 3.3.2. Number of Shoots per Explant

The results indicate that the cotyledon explants cultured on regeneration medium without kanamycin recorded the highest number of shoots (6.5 ± 0.65), followed by the hypocotyl explants (4.75 ± 0.48) on the same medium (Figure 10 and Table 4). The regeneration media containing 50 and 75 mg/L kanamycin yielded 2.5 ± 0.29 and 1 ± 0 shoots per explant, respectively, from the hypocotyl explants (Figure 10 and Table 4). Similar to the percentage of explants forming shoots, the number of shoots per explant also decreased with the increase in kanamycin concentration. Interestingly, no shoot was formed on the cotyledon explants on media with kanamycin. These results suggest that the toxicity of kanamycin is tissue-specific. Hung and Xie [50] observed the tissue-specific toxicity of kanamycin in *Astragalus racemosus*. They found that for the complete inhibition of callus and shoot formation in the cotyledon and hypocotyl, the concentration of kanamycin was the same, but a different concentration was required for root explants. In another study, Sharma et al. [51] found that 50 mg/L kanamycin did not completely inhibit the regeneration of shoots from tomato hypocotyl and cotyledon, but affected them slowly.

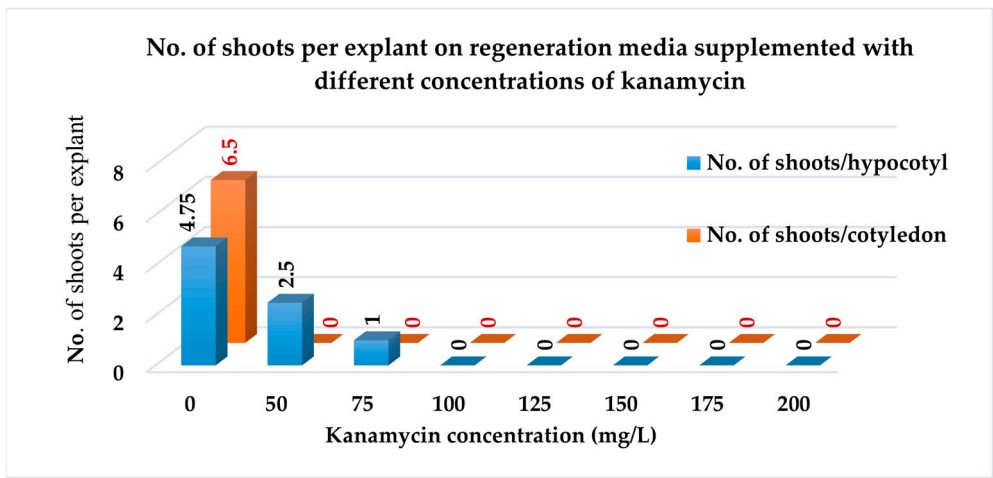

**Figure 10.** Number of shoots per explant on regeneration media supplemented with different concentrations of kanamycin.

### 3.3.3. Rooting and Acclimatization

The highest percentage of root induction was observed in shoots derived from the control regeneration medium (S1) (Figure 11 and Table 5). S1 showed a root formation rate of 72.5% when cultured on the MS medium. All the S1 shoots (100%) produced roots upon being cultured on MS + 0.75 mg/L IAA and MS + 1 mg/L IAA. The percentage of root formation was reduced for S2 (shoots from media containing 50 mg/L kanamycin) and S3 (shoots from media containing 75 mg/L kanamycin) shoots at all concentrations of IAA (Figure 11). The shoot type S3 exhibited the lowest percentage of rooting (10%) in T1, while a similar shoot type in T2 (MS + 0.75mg/L IAA) and T3 (MS + 1.0 mg/L IAA) displayed a rooting percentage of 40%.

The highest number of roots per shoot (32 ± 1.83) was recorded in T2 for S1 shoots, followed by T3 (27.25 ± 1.11) for the same type of shoot. The reduction in the root number followed a similar trend as the percentage of rooting. The number of roots decreased in the shoots derived from media with higher concentrations of kanamycin (Figure 12 and Table 5) and it was lowest for S3 in T1. These results indicate that shoots from media containing a higher concentration of kanamycin partially inhibited rooting in terms of the percentage of shoots inducing roots and the number of roots per shoot. Similarly, a reduction of root formation and root growth with an increase of the kanamycin concentration (>10 mg/L) was observed in Arabidopsis [49]. Of the rooted plants, 80% survived during the acclimatization stage.

**Table 5.** Rooting on shoots derived from kanamycin-treated explants.

| Treatment | % Shoots Forming Roots | No. of Shoots per Explant |
|---|---|---|
| S1 | 90.83 a $\pm$ 4.17 | 20.67 a $\pm$ 3.92 |
| S2 | 75.83 b $\pm$ 7.93 | 8.17 b $\pm$ 1.70 |
| S3 | 30.00 c $\pm$ 4.77 | 5.33 c $\pm$ 1.16 |
| Level of significance | ** | ** |
| MSD value | 7.42 | 1.98 |
| CV (%) | 11.18 | 17.19 |
| T1 | 40.83 b $\pm$ 8.02 | 1.42 c $\pm$ 0.36 |
| T2 | 78.33 a $\pm$ 8.33 | 18.50 a $\pm$ 3.02 |
| T3 | 77.50 a $\pm$ 8.27 | 14.25 b $\pm$ 2.84 |
| Level of significance | ** | ** |
| MSD value | 7.42 | 1.98 |
| CV (%) | 11.18 | 17.19 |
| S1T1 | 72.50 b $\pm$ 4.79 | 2.75 ef $\pm$ 0.48 |
| S1T2 | 100 a $\pm$ 0 | 32.00 a $\pm$ 1.83 |
| S1T3 | 100 a $\pm$ 0 | 27.25 b $\pm$ 1.11 |
| S2T1 | 40.00 c $\pm$ 4.08 | 1.00 f $\pm$ 0.41 |
| S2T2 | 95.00 a $\pm$ 2.89 | 14.25 c $\pm$ 0.85 |
| S2T3 | 92.50 a $\pm$ 4.79 | 9.25 d $\pm$ 1.11 |
| S3T1 | 10.00 d $\pm$ 4.08 | 0.50 f $\pm$ 0.29 |
| S3T2 | 40.00 c $\pm$ 4.08 | 9.25 d $\pm$ 1.11 |
| S3T3 | 40.00 c $\pm$ 4.08 | 6.25 de $\pm$ 0.63 |
| MSD value | 18.39 | 4.39 |
| CV (%) | 11.67 | 16.03 |
| Interaction level (E $\times$ G) significance | ** | ** |

Means in a column that include the same letters are not statistically different at a 5% level using Tukey's HSD test. MSD = minimum significant difference, CV = coefficient of variation, S1 = shoots from from MSB5 + 2 mg/L BAP + 0.5 mg/L IAA, S2 = shoots from MSB5 + 2 mg/L BAP + 0.5 mg/L IAA + 50 mg/L kanamycin, and S3 = MSB5 + 2 mg/L BAP + 0.5 mg/L IAA + 75 mg/L kanamycin. T1 = MS, T2 = 0.75 mg/L IAA, T3 = 1 mg/L IAA. Note: ns—not significant at $p > 0.05$ and **—significant at $p \leq 0.01$; $\pm$—standard error (n = 4); (as ANOVA).

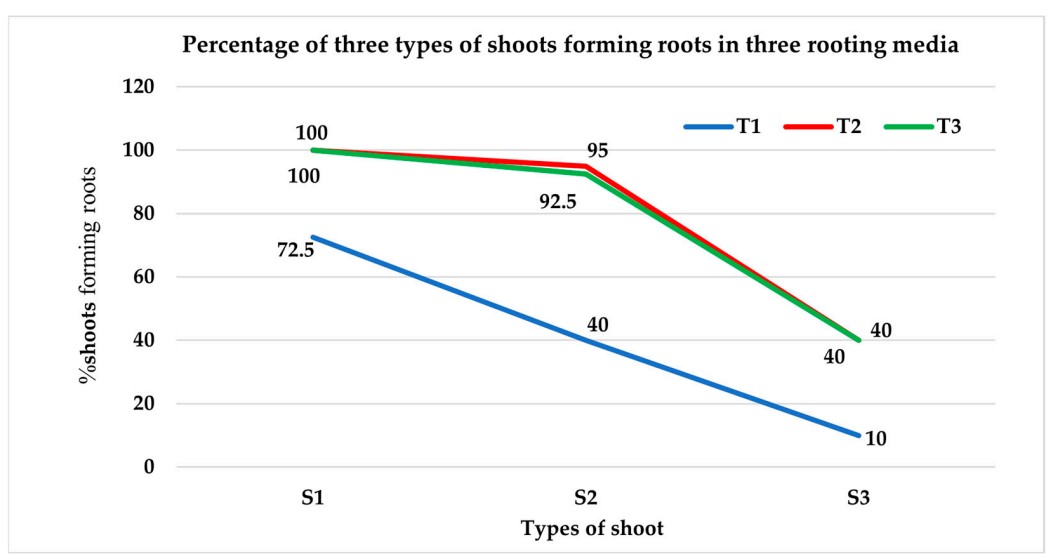

**Figure 11.** Percentage of three types of shoots forming roots on three rooting media.

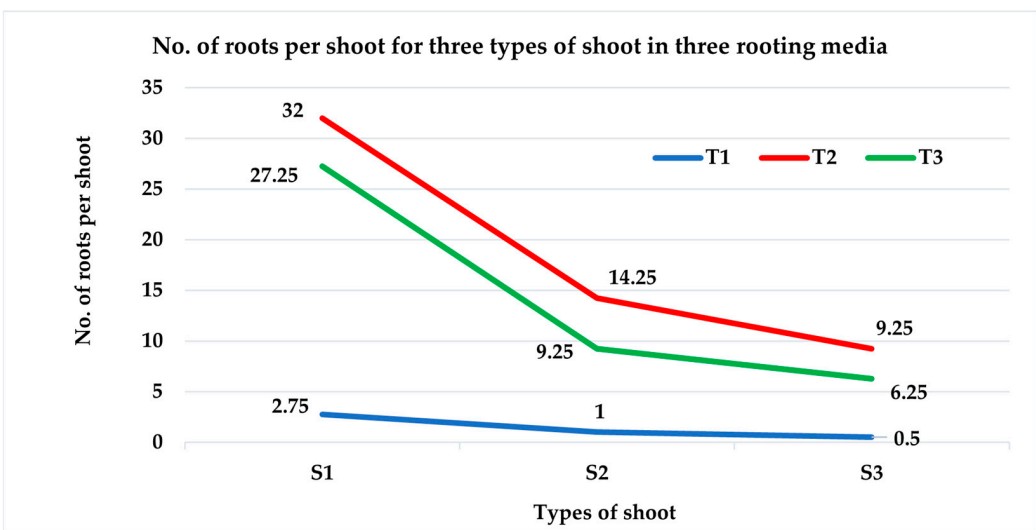

**Figure 12.** Number of roots per shoot for three types of shoots on three rooting media.

## 4. Conclusions

The best regeneration media for hypocotyl and cotyledon explants were found to be MSB5 + 2 mg/L BAP + 0.1 mg/L IAA (T14) and MSB5 + 2 mg/L BAP + 0.5 mg/L IAA (T18), respectively. The shoot elongation appeared to be optimal on MS media without PGRs. Based on the root induction and the survival rate of the regenerants, the optimal concentration of IAA in MS medium for root induction on regenerated shoots was found to be 0.75 mg/L. The findings also indicate that hypocotyl exhibited greater success in promoting a higher percentage of explant-forming shoots, while cotyledon was observed to be more effective in producing a higher number of direct shoot regeneration. The present study only employed BAP and IAA. The omission of other plant growth regulators (PGRs) precludes a comprehensive comparison of the effect of other growth regulators on the regeneration capacity. In future experiments regarding the in vitro regeneration of cv. MT1, it is recommended to include other PGRs and explants, such as the leaf, internode, and shoot tip. This will enable the identification of more optimal combinations of explant and growth regulators for shoot regeneration.

The results of the study indicate that a complete inhibition of shoot regeneration occurred on hypocotyl explants when exposed to medium containing 100 mg/L kanamycin. In contrast, the cotyledon explants exhibited total inhibition of regeneration when subjected to medium containing 50 mg/L kanamycin. The shoots derived from media supplemented with kanamycin showed a decrease in root formation, which was more pronounced in the shoots exposed to a greater concentration of kanamycin. The findings of the present study suggest that 100 mg/L kanamycin is suitable for the screening of transformants derived from hypocotyl of cv. MT1. At an initial concentration of 50 mg/L of kanamycin, the cotyledons were observed to have fully impeded regeneration, while the hypocotyls were still able to produce shoots. However, at a concentration of 75 mg/L, the hypocotyls were not completely inhibited, and a complete inhibiting effect was observed at a concentration of 100 mg/L kanamycin. It is possible that a more accurate concentration within the 0–50 mg/L range may be suitable for cotyledons, while a concentration range of 75–100 mg/L may be appropriate for hypocotyls. Therefore, it is recommended to evaluate smaller intervals of kanamycin concentrations, such as 10, 20, 30, 40, 50, 60, 70, 80, 90, and 100 mg/L, in future studies to obtain more precise outcomes.

**Author Contributions:** Conceptualization, S.A., W.A.S.W.A. and M.A.M.H.; methodology, S.A., W.A.S.W.A., M.A.Y.A. and M.A.M.H.; software, S.A. and R.A.; validation, M.A.M.H., A.S.J. and S.I.I.; formal analysis, S.A. and R.A.; writing—original draft preparation, S.A.; writing—review and editing, S.A., M.A.Y.A. and M.A.M.H.; supervision, M.A.M.H., M.A.Y.A., A.S.J. and S.I.I.; project

administration, M.A.M.H., A.S.J. and S.I.I.; funding acquisition, M.A.M.H. and A.S.J. All authors have read and agreed to the published version of the manuscript.

**Funding:** This research was funded by the Ministry of Higher Education Malaysia (MoHE) through the Fundamental Research Grant Scheme (FRGS/1/2018/STG05/UPM/02/13) and the National Agricultural Technology Programme-Phase-II, Bangladesh (Project ID: 1100001758).

**Data Availability Statement:** The data presented in this study are available upon request from the corresponding author.

**Acknowledgments:** We thank all the staff members of the Agrobiotechnology Laboratory, Department of Agriculture Technology, Faculty of Agriculture, Universiti Putra Malaysia for their technical support during the research. We also extend our gratitude to Farahziatul Roshidah Nazri and Nur Afifah Raban for their help and support. Finally, we thank Maheran Abd Aziz for reviewing the manuscript.

**Conflicts of Interest:** The authors declare no conflict of interest.

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
