# Peer review of "Optimization of In Vitro Regeneration Protocol of Tomato cv. MT1 for Genetic Transformation"

_horticulturae, doi:10.3390/horticulturae9070800_

Round 1
Reviewer 1 Report
The study highlights the variability in the optimal PGR combinations for shoot regeneration and the MIC of kanamycin across the two types of explants. The study is important in several aspects. By continuing to refine and expand the knowledge of tomato transformation techniques, researchers can continually improve the efficiency and success rate of genetic modification, leading to advancements in crop improvement and agricultural sustainability. The work is scientifically sound and can be accepted after minor adjustment.
The followings are the corrections or suggestions that need to be incorporated.
-Abbreviations such as the name of the phytohormone, MS, etc., should be explained when they are first used.
-Justify the novelty of work in Introduction and Discussion.
-Line 32, 66 remove bracket.
- The author uses the combination of BA and IAA. How do you know about this particular combination? What about other cytokinin or auxin combinations?
- Author reported callus formation on shoot elongation media. However, author did not provide any rationale for this observation.
- Add scale bar in each photographs of in vitro regeneration.
- Author used any acclimatizing substrate or directly transferred the in vitro derived plants to the pot having garden soil? For better understanding, author need to add the data of plant survival percentage for in vitro derived plants.
-In its current state, the level of English throughout the manuscript does not meet the journal’s required standard. You may wish to ask a native speaker to check your manuscript for grammar, style and syntax or use the professional language editing options.
In its current state, the level of English throughout the manuscript does not meet the journal’s required standard
Author Response
Point 1: Abbreviations such as the name of the phytohormone, MS, etc., should be explained when they are first used.
Response 1: We thank the reviewer for this suggestion for improving the manuscript. We have now defined the abbreviations when first mentioned in the manuscript.
Point 2: Justify the novelty of work in Introduction and Discussion
Response 2: Although studies on the regeneration protocol of tomatoes are common, it is well-known that crop regeneration and transformation are genotype dependent. In our experiment, we used MT1, a low-land cultivar that is widely grown in Malaysia. This cultivar is also used as a model crop for genetic studies on other crops, (i.e., oil palm). However, in previous studies, the authors only used cotyledons as explants. If other plant parts, such as hypocotyls, could be used for regeneration, it would reduce the number of seeds and more time efficient. Also, a protocol with a lower cost is necessary to facilitate more research. On the other hand, there was a lack of information about the minimum inhibitory concentration (MIC) of kanamycin in non-transformed regenerants for this specific cultivar. These were already mentioned in the introduction chapter. Therefore, in our point of view, using a novel explant (hypocotyl) to improve regeneration efficiency for this specific cultivar and studying the MIC for kanamycin were the novelty of this research.
Point 3: Line 32, 66 remove bracket.
Response 3: Well-spotted. We have removed the bracket in those lines.
Point 4: The author uses the combination of BA and IAA. How do you know about this particular combination? What about other cytokinin or auxin combinations?
Response 4: Previous studies used only cotyledons as explants and zeatin, 2iP, BAP, and IAA as PGRs for evaluating the regeneration efficiency of this particular cultivar MT1. The primary objective of this study was to identify a simple, inexpensive, and time-efficient regeneration protocol for use in tomato transformation experiments. We decided to use combinations of BAP and IAA only analyzing the findings of other authors for other tomato varieties.
Point 5: Author reported callus formation on shoot elongation media. However, author did not provide any rationale for this observation.
Response 5: We appreciate the reviewer’s suggestion. We have added a few lines briefly discussing the potential causes of callus formation on shoot elongation media and supported these with two references that used plain MS media without PGRs for shoot elongation.
Point 6: Add a scale bar in each photograph of in vitro regeneration.
Response 6: We have added a scale bar in each photograph in the revised manuscript.
Point 7: Author used any acclimatizing substrate or directly transferred the in vitro derived plants to the pot having garden soil? For better understanding, author need to add the data of plant survival percentage for in vitro derived plants.
Response 7: In the revised version of materials and methods, we have added information about plant husbandry prior to transfer to the soil and the type of soil used.
Table 3 had a parameter named “survival rate (%). This was actually the survival rate of rooted regenerants on soil, but the explanation was incorrect. To avoid confusion, we have changed the name of the parameter in the table and rewritten the corresponding paragraph as ‘survival rates (%) of rooted shoots on soil’.
Point 8: In its current state, the level of English throughout the manuscript does not meet the journal’s required standard. You may wish to ask a native speaker to check your manuscript for grammar, style and syntax or use the professional language editing options.
Response 8: We thank the reviewer for this suggestion for improving the readability of the manuscript. However, one of the reviewers has recommended minor editing. In fact, the manuscript was reviewed by a person with a good English prior to submission. Nevertheless, we have thoroughly reviewed the manuscript again and improved some sentences to enhance their readability.

Reviewer 2 Report
In the manuscript entitled " Optimization of in vitro regeneration protocol of tomato cv. MT1 for genetic transformation." the authors performed a regeneration optimization that was published on many different other tomato cultivars (also cited in the recent manuscript). The novelty of this manuscript is very low because their results are only slightly different from results on other cultivars. On the other hand, the methods were described in detail, so the experiments can be repeated by others and the presentation of the results is clear, the figures and tables are well-prepared. In general, the manuscript is written in a good manner and order.
I would suggest using different plant growth regulators and different explants to increase the novelty of the research work before publication, as mentioned in the Conclusion part.
Author Response
Point 1: In the manuscript entitled " Optimization of in vitro regeneration protocol of tomato cv. MT1 for genetic transformation." the authors performed a regeneration optimization that was published on many different other tomato cultivars (also cited in the recent manuscript). The novelty of this manuscript is very low because their results are only slightly different from results on other cultivars.
Response 1: Although studies on the regeneration protocol of tomatoes are common, it is well-known that crop regeneration and transformation are genotype dependent. In our experiment, we used MT1, a low-land cultivar that is widely grown in Malaysia. This cultivar is also used as a model crop for genetic studies on other crops, (i.e., oil palm). However, in previous studies, the authors only used cotyledons as explants. If other plant parts, such as hypocotyls, could be used for regeneration, it would reduce the number of seeds and more time efficient. Also, a protocol with a lower cost is necessary to facilitate more research. On the other hand, there was a lack of information about the minimum inhibitory concentration (MIC) of kanamycin in non-transformed regenerants for this specific cultivar. These were already mentioned in the introduction chapter. Therefore, in our point of view, using a novel explant (hypocotyl) to improve regeneration efficiency for this specific cultivar and studying the MIC for kanamycin were the novelty of this research.
Point 2: I would suggest using different plant growth regulators and different explants to increase the novelty of the research work before publication, as mentioned in the Conclusion part.
Response 2: We thank the reviewer for this suggestion for enhancing the novelty of the manuscript. However, we are unable to conduct additional experiments as suggested due to time constraints and limited resources. We believe that our current findings have significant value in advancing the existing regeneration protocol of this particular cultivar by using a novel explant, hypocotyl.

Reviewer 3 Report
Authors need to check scientific unit writing. Be consistent in all places. In text, Liter was spelled as mg/L but some places and in figure 10 was written as mg/l
Line 60: Define BAP
Line 63-73: Rephrase this paragraph to make the sentences more connected to each other.
Line 83-92: Separate number and measurement units
Line 86: Tween-20
Line 88: Define MS medium (even though some readers know what is MS)
Line 83: cultivar or variety? Better to maintain tomato cv. MT1
Line 101-102: Justify why different concentrations of IAA were used for roots
Line 109: Table 1. Naming treatment from T1-T20 will only confuse the readers. Every time readers need to check what is T1, T2 etc. Suggest removing the name/code in the text. Can maintain the name/code in Tables and Figures.
Line 148-161: any reference for collecting data?
Line 169-179: Authors still need to state the treatment name/code here while also stating the treatment combination. Refer comment before
Line 184-185: I do think that author still needs to mention the author's name before putting the [x] behind it. Better refer to previous papers or instructions to authors
Minor editing and spell check is required
Author Response
Point 1: Authors need to check scientific unit writing. Be consistent in all places. In text, Liter was spelled as mg/L but some places and in figure 10 was written as mg/l.
Response 1: We thank reviewer for this suggestion to improve the manuscript. We have corrected the inconsistencies throughout the manuscript including the figures.
Point 2: Line 60: Define BAP
Response 2: We have defined the BAP.
Point 3: Line 63-73: Rephrase this paragraph to make the sentences more connected to each other.
Response 3: We have improved the paragraph accordingly.
Point 4: Line 83-92: Separate number and measurement units.
Response 4: We have separated the number and measurement units as suggested.
Point 5: Line 86: Tween-20
Response 5: We have made the correction.
Point 6: Line 88: Define MS medium (even though some readers know what is MS)
Response 6: We have defined the abbreviation.
Point 7: Line 83: cultivar or variety? Better to maintain tomato cv. MT1
Response 7: We have corrected the inconsistency of the term.
Point 8: Line 101-102: Justify why different concentrations of IAA were used for roots.
Response 8: We have mentioned the justification of using different IAA concentrations in the objective and materials and methods of the study.
Point 9: Table 1. Naming treatment from T1-T20 will only confuse the readers. Every time readers need to check what is T1, T2 etc. Suggest removing the name/code in the text. Can maintain the name/code in Tables and Figures.
Response 9: Since the various combinations of PGRs and explants were difficult to read at a glance, abbreviated forms were used. We have elaborated the treatments in the text as suggested by the reviewer.
Point 10: Line 148-161: any reference for collecting data?
Response 10: We have added a reference for data collection.
Point 11: Line 169-179: Authors still need to state the treatment name/code here while also stating the treatment combination. Refer comment before.
Response 11: We have added the factors and treatments in the text.
Point 12: Line 184-185: I do think that author still needs to mention the author's name before putting the [x] behind it. Better refer to previous papers or instructions to authors
Response 12: Good point. We have added the author’s name throughout the revised manuscript.

Round 2
Reviewer 2 Report
The authors convinced me with their answers, that the novelty of this research is the used protocols only on the MT1 cultivar. Otherwise, the manuscript is written in a good manner and order. However, a minor correction I would suggest. In Figure 4 the scalebar is the same in all the 3 photos, however, the zoom is visibly different (the size of the glass tubes is different). Please, correct the size of the scale bars.